# K-DeCore: Facilitating Knowledge Transfer in Continual Structured Knowledge Reasoning via Knowledge Decoupling

**Yongrui Chen**[1,2]    **Yi Huang**[3,4,∗]    **Yunchang Liu**[1,2]    **Shenyu Zhang**[1,2]
**Junhao He**[1,2]    **Tongtong Wu**[5]    **Guilin Qi**[1,2]    **Tianxing Wu**[1,2,∗]

[1]School of Computer Science and Engineering, Southeast University, China
[2]Key Laboratory of New Generation Artificial Intelligence Technology and
its Interdisciplinary Applications (Southeast University), Ministry of Education, China
[3]JIUTIAN Research, China Mobile
[4]Department of Computer Science and Technology, Tsinghua University, China
[5]Department of Data Science & AI, Monash University, Australia
{yongruichen, tianxingwu}@seu.edu.cn, huangyi@chinamobile.com

## Abstract

Continual Structured Knowledge Reasoning (CSKR) focuses on training models to handle sequential tasks, where each task involves translating natural language questions into structured queries grounded in structured knowledge. Existing general continual learning approaches face significant challenges when applied to this task, including poor generalization to heterogeneous structured knowledge and inefficient reasoning due to parameter growth as tasks increase. To address these limitations, we propose a novel CSKR framework, K-DeCore, which operates with a fixed number of tunable parameters. Unlike prior methods, K-DeCore introduces a knowledge decoupling mechanism that disentangles the reasoning process into task-specific and task-agnostic stages, effectively bridging the gaps across diverse tasks. Building on this foundation, K-DeCore integrates a dual-perspective memory consolidation mechanism for distinct stages and introduces a structure-guided pseudo-data synthesis strategy to further enhance the model's generalization capabilities. Extensive experiments on four benchmark datasets demonstrate the superiority of K-DeCore over existing continual learning methods across multiple metrics, leveraging various backbone large language models.

## 1 Introduction

Structured Knowledge Reasoning (SKR) aims to translate the natural language question into structured queries over discrete, structured knowledge —such as relational databases, knowledge graphs, and dialogue states. It is essential for a wide range of real-world applications, including legal judgment prediction [1], clinical decision support [2, 3], and financial analysis [4]. Recent advances in Large Language Models (LLMs) have demonstrated impressive capabilities for structured reasoning when provided with appropriate prompting or fine-tuning [5, 6, 7].

However, most existing methods operate under a static assumption that SKR tasks are singular in type and remain unchanged throughout both the training and deployment phases. This assumption is misaligned with real-world scenarios, where models must continuously adapt to new reasoning tasks across various forms of structured knowledge. For instance, a virtual assistant like Siri or Alexa must

---

∗Corresponding authors.

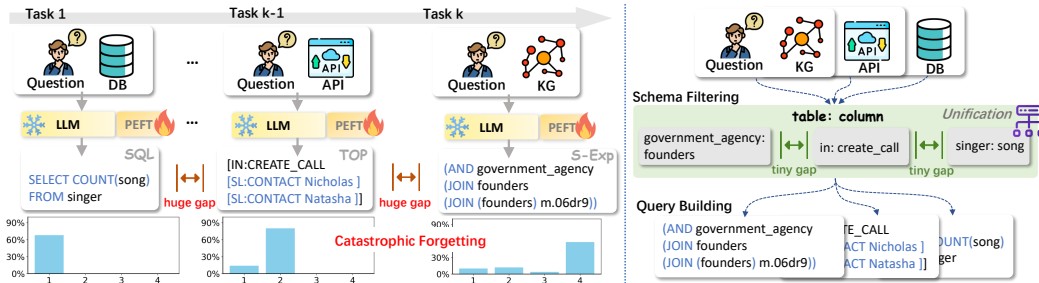

Figure 1: a) Overview of the continual SKR task. The backbone LLM is frozen and only the PEFT module if tunable. b) SKR based on knowledge decoupling. Schema filtering refines the scope of a given schema and impacts various SKR performance. The differences between schema filtering in these SKR tasks are minimal, making it potential for *knowledge transfer*.

dynamically adjust its knowledge base to handle a variety of tasks such as managing calendar events, to-do lists, and smart home controls, each characterized by distinct schemas and query languages. Therefore, it is essential to empower the model to leverage knowledge transfer across different tasks during the continual learning process [8].

Recent works to continual SKR have primarily focused on designing the training strategy for a single type of structured knowledge (e.g., continual text-to-SQL) [9, 10] or employing parameter-efficient fine-tuning (PEFT) techniques to allocate task-specific parameters [11, 12, 13]. However, these methods often fail to generalize across heterogeneous structured knowledge and suffer from parameter growth proportional to the number of tasks, ultimately hindering reasoning efficiency.

We observe that schema filtering, i.e., identifying schema elements relevant to query construction, (Figure 1(b)) is a shared, reusable component across diverse SKR tasks and significantly influences overall performance. We hypothesize that decoupling pattern filtering from downstream stages and unifying its input-output format across tasks can enhance robustness to task-specific variations. This consistency facilitates knowledge transfer and improves SKR performance without requiring parameter growth or task-specific reviews. For query construction stages with high task variance, their contribution to SKR depends on the structural richness of replayed queries. We propose to automatically synthesize structurally diverse replay samples, enabling more informative replay memory usage within the same capacity budget.

In this paper, we propose K-DECORE (Knowledge DEcoupling for COntinual REasoning), a novel continual SKR framework. K-DECORE comprises a backbone LLM and two lightweight PEFT modules: a schema filter and a query builder, responsible for capturing task-specific and task-agnostic knowledge, respectively. This decoupled design fosters forward and backward knowledge transfer. To mitigate the model's tendency to forget previously learned tasks, K-DECORE incorporates a dual-perspective memory mechanism (Section 3.2) designed to retain replay samples from both schema and query structure viewpoints. Specifically, the schema memory captures representative schema instances, ensuring the filter maintains comprehensive coverage across tasks, while the query memory emphasizes preserving diverse query structures inherent to SKR tasks. Furthermore, to enhance the model's generalization to unseen patterns, we introduce a query synthesis strategy that generates novel structured queries for replay. We evaluate K-DECORE on task streams comprising four diverse SKR tasks, where it consistently outperforms strong baselines and achieves state-of-the-art results across multiple metrics. In summary, the contributions of this paper include:

- We propose a novel continual structured knowledge reasoning framework, leveraging knowledge decoupling to enable effective knowledge transfer across diverse tasks. To the best of our knowledge, this is the first work to explore continual learning across heterogeneous SKR tasks.

- We propose a dual-perspective memory construction mechanism that specifically synthesizes pseudo queries for novel structures, with the goal of enhancing the LLM's generalization capability.

- We curate task streams from four SKR benchmarks and conduct comprehensive experiments. Empirical results demonstrate that K-DECORE consistently surpasses strong baselines across multiple evaluation metrics, establishing its effectiveness in continual structured knowlegde reasoning.

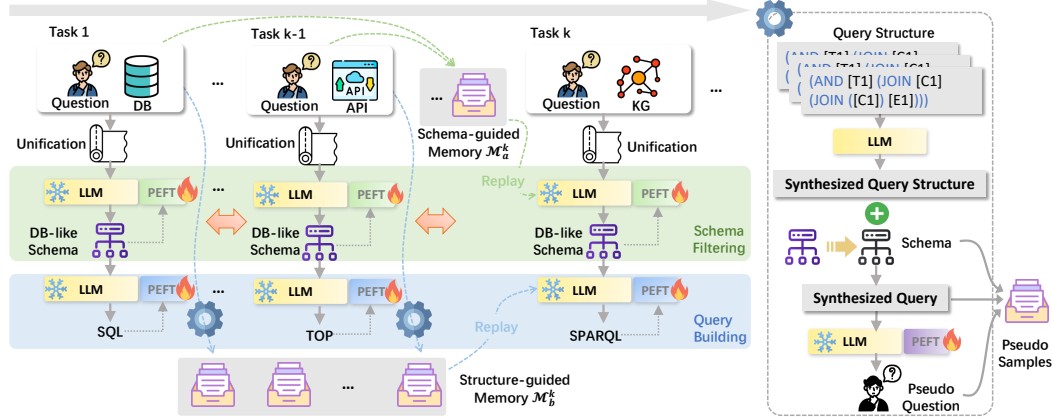

Figure 2: The left panel presents the K-DeCore training framework, organized into two key stages for each SKR task: schema filtering and query building, each supported by specialized PEFT modules. By unifying the schema, the framework aims to bridge the gap between tasks, effectively enabling the *knowledge transfer*. The right panel illustrates the creation process of structure-guided synthetic pseudo samples, designed to offer a more structurally diverse set of examples.

## 2  Preliminary

**Structured Knowledge Reasoning** Given a natural language question $\mathcal{Q}$ and accessible structured knowledge $\mathcal{S}$, such as a database, a dialogue state, or a knowledge graph, the goal of Structured Knowledge Reasoning is to generate a structured query $\mathcal{Y}$. Formally, this can be represented as: $\mathcal{Y} = f_\theta(\mathcal{Q}, \mathcal{S})$, where $f_\theta$ denotes a LLM-based reasoner parameterized by $\theta$.

In this paper, we focus on three commonly used types of structured knowledge: a) A *database* is represented as $(\mathcal{C}, \mathcal{T})$, where $\mathcal{C} = c_1, \ldots, c_n$ denotes the column names and $\mathcal{T} = \{t_1, \ldots, t_m\}$ denotes the table names. b) A *knowledge graph* is generally composed of subject-predicate-object triples, expressed as $\{\langle s, p, o \rangle \,| s \in \mathcal{E}, p \in \mathcal{R}, o \in \mathcal{E} \cup \Gamma\}$, where $\mathcal{E}$ is the set of entities, $\mathcal{R}$ is the set of relations, and $\Gamma$ is the set of types. c) A *dialogue state* is represented as a collection of predefined intents paired with their corresponding slots, indicated by $\{\langle \mathcal{I}_1, s_1 \rangle, \ldots, \langle \mathcal{I}_n, s_n \rangle\}$, where each $\mathcal{I}_i$ represents an intention and $s_i$ denotes the associated slots.

**Problem Formulation** Let $f_\theta$ be trained sequentially on $K$ SKR tasks, denoted by $\{\mathcal{D}^1, \mathcal{D}^2, \ldots, \mathcal{D}^K\}$. Each task $\mathcal{D}^i$ consists of a training set and a test set: $\mathcal{D}^k = \mathcal{D}^k_{\text{train}} \cup \mathcal{D}^k_{\text{test}} = \{(\mathcal{Q}_i, \mathcal{S}_i, \mathcal{Y}_i)\}_{i=1}$. To encourage $f_\theta$ to continually acquire reasoning capabilities over diverse types of structured knowledge, for tasks $\mathcal{D}^k$ and $\mathcal{D}^j$ ($k \neq j$), the corresponding structured knowledge $\mathcal{S}(\mathcal{D}^k)$ and $\mathcal{S}(\mathcal{D}^j)$ are of different types, i.e., $\text{Type}(\mathcal{S}(\mathcal{D}^k)) \neq \text{Type}(\mathcal{S}(\mathcal{D}^j))$. Our objective is to ensure that $f_\theta$ performs well on each $\mathcal{D}^k_{\text{test}}$ after sequentially learning on all $\mathcal{D}^k_{\text{train}}$, for all $k \in \{1, \ldots, K\}$.

## 3  K-DeCore

Figure 2 illustrates the overall architecture of the proposed K-DeCore. It is built upon a backbone LLM $f$ with parameter $\theta$, and incorporates three PEFT modules: $\mathbf{P}_a$, $\mathbf{P}_b$, and $\mathbf{P}_c$. Throughout training, the backbone parameters $\theta$ remain fixed; only the parameters of the PEFT modules are updated, enabling efficient adaptation to the task while preserving the pretrained knowledge of the LLM. The entire method can be divided into two parts: *continual knowledge decoupling* (Section 3.1) and *dual-perspective memory construction* (Section 3.2), which are described in detail below.

### 3.1  Continual Knowledge Decoupling

Unlike traditional continual learning methods [13, 14] in general fields, our proposed K-DeCore innovatively bifurcates the reasoning process over the structured knowledge into two distinct yet synergistic phases: *schema filtering* and *query building*. The schema filtering phase is designed to distill the essential schema components $\mathcal{S}^* \subseteq \mathcal{S}$ necessary for constructing the final query $\mathcal{Y}$ from

the initial schema $\mathcal{S}$. Subsequently, the query building phase focuses on generating the query $\mathcal{Y}$ using both the original schema $\mathcal{S}$ and the refined schema $\mathcal{S}^*$ obtained from schema filtering. This decoupling strategy offers dual advantages: firstly, within-task pattern filtering has been empirically demonstrated to enhance the performance of query generation by reducing the search space; secondly, the relatively stable output format of schema filtering across diverse SKR tasks enables the model to facilitate more robust knowledge transfer from preceding tasks during the schema filtering phase. In the following sections, we will delve into a detailed exploration of these two components.

### 3.1.1 Task-agnostic Continual Schema Filtering

K-DECORE trains an independent PEFT module, denoted as $\mathbf{P}_a$, for the schema filtering stage across all tasks. This module is integrated with the backbone LLM $f_\theta$ to predict the relevant schema set $\mathcal{S}^*$, i.e., $\mathcal{S}^* = f(\mathcal{Q}, \mathcal{S}; \theta, \mathbf{P}_a)$. During the training process for task $k$, the module $\mathbf{P}_a^k$ is initialized with parameters from the checkpoint of the preceding task, represented as $\mathbf{P}_a^{k-1}$. To ensure that $\mathbf{P}_a$ perceives a consistent style akin to previous tasks while learning new SKR tasks, thereby mitigating forgetting due to training, the knowledge schema $\mathcal{S}$ from various SKR tasks is standardized into a database-like (DB-like) *unified schema representation*. This standardization further reduces task-specific discrepancies and enhances effective knowledge transfer. The DB style is chosen as the unified format for two primary reasons: (a) it is likely the most prevalent form of $\mathcal{S}$ and is familiar to LLMs, and (b) its straightforward relational structure simplifies the conversion of other schema types. Concretely, for each $(\mathcal{Q}, \mathcal{S}, \mathcal{Y})$, $\mathcal{S}$ is first converted to $\Omega = (\Phi, \Psi)$, where $\Phi = \{\phi_1, \ldots, \phi_{|\Phi|}\}$ denotes the set of abstract table names, and $\Psi = \{\psi_1, \ldots, \psi_{|\Psi|}\}$ denotes the set of abstract column names. Then, $f(\theta, \mathbf{P}_a)$ predicts the useful schema $\Omega^*$ from $\Omega$, i.e., $\Omega^* = f(\mathcal{Q}, \mathcal{Z}; \theta, \mathbf{P}_a)$, where $\widetilde{\Omega}$ denotes the textual representation of $\Omega$:

$$\widetilde{\Omega} = \phi_1 : \psi_1^1, \ldots, \psi_1^n \mid \phi_2 : \psi_2^1, \ldots, \psi_2^n \mid \ldots \mid \phi_m : \psi_m^1, \ldots, \psi_m^n. \tag{1}$$

During training, the following loss function is minimized to optimize $\mathbf{P}_a$, while keeping $\theta$ frozen:

$$\mathcal{L}(\mathcal{D}^k; \theta, \mathbf{P}_a) = -\sum_{i=1}^{|\mathcal{D}^k|} \sum_{j=1}^{|\widetilde{\Omega}_i^*|} \log P(\omega_j^* \mid \mathcal{Q}_i, \widetilde{\Omega}_i, \omega_{<j}^*; \theta, \mathbf{P}_a) + \sum_{k'=1}^{k-1} \mathcal{L}(\mathcal{M}_a^{k'}; \theta, \mathbf{P}_a), \tag{2}$$

where $P(z_j^* \mid \mathcal{Q}_i, \widetilde{\Omega}_i, \omega_{<j}^*; \theta, \mathbf{P}_a)$ denotes the probability of each token $z_j^*$ generated by autoregression. To further mitigate the model's forgetting, we incorporate a review loss $\mathcal{L}(\mathcal{M}_a^{k'}; \theta, \mathbf{P}_a)$ computed on the memory $\mathcal{M}_a^{k'}$ from task $k'$, which will be detailed in Section 3.2.

**How to convert to DB style?** In this paper, various forms of structured knowledge can be easily transformed into $\bar{\mathcal{S}}$. For a DB $(\mathcal{T}, \mathcal{C})$, each table name $t_i \in \mathcal{T}$ is mapped to $\phi$, while each column name $c_i \in \mathcal{C}$ is mapped to $\psi$. For a KG subgraph $\{\langle s, p, o \rangle \mid s \in \mathcal{E}, p \in \mathcal{R}, o \in \mathcal{E} \cup \Gamma\}$, each entity type $\gamma \in \Gamma$ and each relation name $t_i \in \mathcal{T}$ are treated as $\phi$, while each relation or property $r \in \mathcal{R}$ and each column name $c_i \in \mathcal{C}$ are treated as $\psi$. For a dialogue state $\{\langle \mathcal{I}_1, s_1 \rangle, \ldots, \langle \mathcal{I}_n, s_n \rangle\}$, each intention $\mathcal{I}_i$ is considered as $\phi$, and each slot name $s_i$ is considered as $\psi$. See Appendix A for details.

### 3.1.2 Task-specific Continual Query Building

Similar to schema filtering, K-DECORE trains another separate PEFT module, denoted as $\mathbf{P}_b$, specifically for the query building phase across all tasks. To avoid error propagation within the pipeline, we ensure that $f(\theta, \mathbf{P}_b)$ also considers the complete schema $\mathcal{S}$ during reasoning, expressed as $\mathcal{Y} = f(\mathcal{Q}, \mathcal{S}^*, \mathcal{S}; \theta, \mathbf{P}_b)$. At this stage, the primary focus of the function $f(\theta, \mathbf{P}_b)$ is to capture the semantics of the problem and the logical and structural features of queries associated with various SKR tasks during reasoning. As shown in Figure 1, the query structures for different SKR tasks are highly task-specific and can vary significantly, which heightens the risk of forgetting at this stage. To address this risk, we include a small number of query samples from previous tasks during training. The following loss function is minimized to optimize $\mathbf{P}_b$:

$$\mathcal{L}(\mathcal{D}^k; \theta, \mathbf{P}_b) = -\sum_{i=1}^{|\mathcal{D}^k|} \sum_{j=1}^{|\mathcal{Y}_i|} \log P(y_j | \mathcal{Q}_i, \mathcal{S}_i^*, \mathcal{S}_i, y_{<j}; \theta, \mathbf{P}_b) + \sum_{k'=1}^{k-1} \mathcal{L}(\mathcal{M}_b^{k'}; \theta, \mathbf{P}_b), \tag{3}$$

where $y_j$ denotes the $j$-th token of the target query $\mathcal{Y}_i$ and $\mathcal{M}_b^{k'}$ represents the memory dedicated to storing structural information of queries, which will be elaborated on in Section 3.2. Notably, here we directly use the raw schema format $\mathcal{S}$ to enable $f(\theta, \mathbf{P}_b)$ to generate the executable queries.

---

**Algorithm 1** Structure-guided Query Synthesis

---

**Require:** The training set of the $k$-th task, represented as $\mathcal{D}_{\text{train}}^k = \{(\mathcal{Q}_i, \mathcal{S}_i, \mathcal{Y}_i)\}_{i=1}^{N^k}$; The set of all query structures present in $\mathcal{D}_{\text{train}}^k$ is denoted by $\mathcal{M}_\mathcal{Y} = \{\mathcal{Y}_1^*, \ldots, \mathcal{Y}_{N_1}^*\}$, while the set of all available schema sets is denoted by $\mathcal{M}_\mathcal{S} = \{\mathcal{S}_1, \ldots, \mathcal{S}_{N_2}\}$.

1: $\mathcal{M}_{\text{pseudo}}^k \leftarrow \emptyset$
2: **for** $|\mathcal{M}_{\text{pseudo}}^k| < \mathcal{N}$ **do**
3:   $\mathcal{S}^+ \leftarrow \text{RANDOM}(\mathcal{M}_\mathcal{S}, 1), \mathcal{Y}^* \leftarrow \text{RANDOM}(\mathcal{M}_\mathcal{Y}, T)$   ▷ Sampling schema and structures.
4:   $\mathcal{Y}_{\text{pseudo}}^* \leftarrow f(\mathcal{Y}^*; \theta)$          ▷ Synthesize a pseudo query structure.
5:   $\mathcal{Y}^+ \leftarrow \text{FILLSCHEMA}(\mathcal{Y}_{\text{pseudo}}^*, \mathcal{S}^+)$     ▷ Fill the schema into the slots of the structure.
6:   **if** $\text{EXECUTE}(\mathcal{Y}^+, \mathcal{S}^+) = \texttt{"success"}$ **then**
7:    $\mathcal{Q}^+ \leftarrow f(\mathcal{Y}^+, \mathcal{S}^+; \theta, \mathbf{P}_c)$     ▷ Generate a pseudo natural language question.
8:    $\mathcal{M}_{\text{pseudo}}^k \leftarrow \mathcal{M}_{\text{pseudo}}^k \cup \{(\mathcal{Q}^+, \mathcal{S}^+, \mathcal{Y}^+)\}$
9:   **end if**
10: **end for**
11: **return** $\mathcal{M}_{\text{pseudo}}^k$

---

## 3.2 Dual-perspective Memory Construction

**Schema-guided Memory Sampling.** The goal of this sampling strategy is to construct memory $\mathcal{M}_a^k$ for each task $\mathcal{D}^k$ to relieve $f(\theta, \mathbf{P}_a)$ from the burden of memorizing schema filtering knowledge. Since we have unified the representation of structured knowledge and tasks, the primary gap between different tasks lies in the domain of their schemas. Therefore, it is essential to select samples with **representative schemas**. Formally, for each training sample $\mathcal{X} = (\mathcal{Q}, \mathcal{S}, \mathcal{Y}) \in \mathcal{D}_{\text{train}}^k$, the process begins with the extraction of the relevant schema set $\mathcal{S}^*$ from $\mathcal{Y}$. Subsequently, all samples in $\mathcal{D}_{\text{train}}^k$ are partitioned into $\mathcal{N}$ clusters. The distance between two samples $\mathcal{X}_1$ and $\mathcal{X}_2$ is defined by the cosine similarity $d_1(\mathcal{X}_1, \mathcal{X}_2) = \cos\big(g(\mathcal{S}_1^*), g(\mathcal{S}_2^*)\big)$, where $g$ denotes an encoder-only LLM. Finally, the sample closest to the center of each cluster is selected, and $(\mathcal{Q}, \mathcal{S}, \mathcal{S}^*)$ is added to $\mathcal{M}_a^k$.

**Structure-guided Memory Construction.** This strategy aims to preserve the sample set $\mathcal{M}_b^k$ with diverse query structure features, which are utilized for the rehearsal process of $f(\theta, \mathbf{P}_b)$. The memory $\mathcal{M}_b^k$ consists of two parts, denoted as $\mathcal{M}_b^k = \mathcal{M}_{\text{real}}^k \cup \mathcal{M}_{\text{pseudo}}^k$. First, $\mathcal{M}_{\text{real}}^k$ comprises representative samples selected from $\mathcal{D}_{\text{train}}^k$. This selection process is analogous to the one used for constructing $\mathcal{M}_a^k$, with the main difference being that the sample distance within a cluster is defined by $d_2(\mathcal{X}_1, \mathcal{X}_2) = \cos\big(g(\mathcal{Y}_1^*), g(\mathcal{Y}_2^*)\big)$. Here, $\mathcal{Y}^*$ denotes the structure of the query $\mathcal{Y}$, which is formed by substituting the schema elements with placeholders. The structure of the s-expression in Figure 1 is represented as (AND [T1] (JOIN [C1] (JOIN ([C1]) [E1]))), where [T1], [C1], and [E1] serve as placeholders for an abstract table, an abstract column, and an entity, respectively.

Unlike existing methods [9, 10] which primarily preserve training-time structures (like $\mathcal{M}_{\text{real}}^k$) seen by $f_\theta$, our K-DECORE employs $\mathcal{M}_{\text{pseudo}}^k$ to boost zero-shot reasoning by introducing novel structures absent from $\mathcal{D}_{\text{train}}^k$. This is achieved through a structure synthesis process, detailed in Algorithm 1. The procedure begins by identifying the repertoire of query structures present in $\mathcal{D}_{\text{train}}^k$, denoted as $\mathcal{M}_\mathcal{Y} = \{\mathcal{Y}_1^*, \ldots, \mathcal{Y}_{N_1}^*\}$, and the available schema sets, $\mathcal{M}_\mathcal{S} = \{\mathcal{S}_1^*, \ldots, \mathcal{S}_{N_2}^*\}$. Iteratively, our backbone model $f_\theta$ is leveraged to generate novel structures: $T$ distinct structures $\mathcal{Y}^* \in \mathcal{M}_\mathcal{Y}$ are sampled and synthesized into a new structure $\mathcal{Y}_{\text{pseudo}}^*$ guided by carefully curated demonstrations. Here $\mathcal{Y}_{\text{pseudo}}^*$ can be complex SQL queries with nested subqueries or S-expressions featuring multi-step compositional reasoning; detailed examples and prompts of these synthesized structures can be found in the Appendix B.3. This query structure $\mathcal{Y}_{\text{pseudo}}^*$ is then instantiated into a concrete query $\mathcal{Y}_{\text{pseudo}}$ by populating it with a randomly sampled schema $\mathcal{S}^* \in \mathcal{M}_\mathcal{S}$. Only generated queries $\mathcal{Y}_{\text{pseudo}}$ that execute correctly, thereby ensuring legality and semantic validity, are retained. Finally, a PEFT module, denoted as $\mathbf{P}_c$, designed to generate the pseudo NLQ $\mathcal{Q}^+$ and is trained by optimizing

$$\mathcal{L}(\mathcal{D}^k; \theta, \mathbf{P}_c) = -\sum_{i=1}^{|\mathcal{D}^k|} \sum_{j=1}^{|\mathcal{Q}_i|} \log P(q_j | \mathcal{Y}, q_{<j}; \theta, \mathbf{P}_c), \tag{4}$$

where $q_j$ denotes the $j$-th token of real $\mathcal{Q}$. Notably, $\mathbf{P}_c^k$ is tailored specifically for each task $\mathcal{D}^k$. Once $\mathcal{M}_b^k$ is constructed, $\mathbf{P}_c^k$ serves as the initialization for training $\mathbf{P}_c^{k+1}$. Since $\mathbf{P}_c^k$ is constructed only once per task, it does not introduce the forgetting problem. To construct $\mathcal{M}_b^k$, we combined the real memory $\mathcal{M}_{\text{real}}^k$ with the synthetic memory $\mathcal{M}_{\text{pseudo}}^k$ in varying proportions. In our experiments, we systematically investigated different synthetic sample percentage to identify the optimal balance.

# 4 Experiments

## 4.1 Experimental Setup

**Datasets.** We conduct experiments on four widely-used SKR datasets, covering DB, KG, DS reasoning. **Spider** [15] is a DB reasoning benchmark where each NLQ is translated into a complex SQL query. These queries often require multi-table `JOIN` operations, `GROUP BY` clauses, and nested subqueries, demanding sophisticated database reasoning capabilities. **ComplexWebQuestions (CWQ)** [16] is a KG reasoning dataset where each NLQ corresponds to an executable SPARQL query. It involves up to 4-hop reasoning over a knowledge graph with complex constraints such as comparisons, aggregations, and nested conditions. **GrailQA** [17] is a KG reasoning benchmark featuring NLQs that require complex multi-hop reasoning to build s-expressions over the Freebase knowledge graph. **MTOP** [18] is designed for multilingual semantic parsing in task-oriented dialogue systems. Each NLQ is parsed into a TOP representation, modeling nested intents and slots.

As detailed in Section 2, we utilize the aforementioned four datasets to construct three distinct sequential task streams, where each task $\mathcal{D}_{\text{train}}$ comprises a single dataset. The input and output spaces vary across different tasks, as illustrated in Figure 1. Additionally, to emulate scenarios with limited training data, each individual task $\mathcal{D}_{\text{train}}^k$ is constrained to $|\mathcal{D}_{\text{train}}^k| = 1000$ and $|\mathcal{D}_{\text{test}}^i| = 300$.

**Evaluation Metrics.** In line with existing studies [13, 12], we employ three metrics to evaluate the performance of the methods: a) $\text{AA}_a = \frac{1}{K}\sum_{k=1}^{K} acc_{k,K}$, which assesses the overall accuracy across all tasks; b) $\text{BWT} = \frac{1}{K-1}\sum_{k=1}^{K-1}(acc_{k,K} - acc_{k,k})$, which measures the extent of forgetting knowledge from previous tasks; c) $\text{FWT} = \frac{1}{K-1}\sum_{k=2}^{K}(acc_{k,k} - acc_{k,0})$, which evaluates the ability to forward transfer knowledge from past tasks to new ones. Here, $acc_{k,j}$ represents the test accuracy on $\mathcal{D}_{\text{test}}^k$ after training on $\mathcal{D}^j$, and $acc_{k,0}$ denotes the test accuracy on $\mathcal{D}_{\text{test}}^k$ only training on $\mathcal{D}_{\text{train}}^k$.

**Compared Methods.** We conduct a comprehensive comparison against the following three types of baselines: **a) FINE-TUNING**: Represents the performance of a standard, vanilla model without any continual learning enhancements. **b) Rehearsal-based Methods:** Includes methods such as EMAR [19] and SFNET [9], which require replaying real historical examples or utilizing additional unlabeled data. **c) PEFT-based Methods:** Covers techniques like PROGPROMPT[14], O-LORA [11], C3 [12], and SAPT [13]. Additionally, we establish an upper-bound baseline, Multi-Task, where for each task $\mathcal{D}^k$, the model $f_\theta$ is trained jointly on all data of $\mathcal{D}_{\text{train}}^{(0:k)}$.

**Backbone LLMs.** To thoroughly assess the performance across different backbone models, we employed three widely used LLMs of varying sizes: 1) T5-LARGE, 2) LLAMA3-8B-INSTRUCT, and 3) QWEN2.5-7B-INSTRUCT. Most existing methods in the literature are tailored for encoder-decoder architectures, such as T5, which inherently limits their applicability to the current generation of LLMs, the majority of which adopt a decoder-only design. Consequently, when employing LLaMA3 and QWEN2.5 as the backbone, these methods are excluded from consideration due to architectural incompatibility. This distinction underscores the novelty of our approach, as it represents the first systematic exploration of leveraging decoder-only LLMs for the continual SKR task.

**Implementation Details.** Our experiments were conducted using a single NVIDIA RTX 4090 GPU. The hyperparameters were configured as follows (see Appendix C for further details): a) We employed LoRA [20] as the PEFT module. b) The memory size for each task was set to $|\mathcal{M}_a^k| = 5$ and $|\mathcal{M}_b^k| = 5$. c) The ratio of real to pseudo memory samples, $|\mathcal{M}_{\text{real}}^k|$ to $|\mathcal{M}_{\text{pseudo}}^k|$, was maintained at $4:1$. d) We used a batch size of 12 and set the learning rate to $5 \times 10^{-5}$. e) The number of training epochs was fixed at 5. f) $T$ in Algorithm 1 is set to 5. g) The backbone LLM for pseudo samples synthesis is set to QWEN2.5-7B-INSTRUCT. All of our data and codes are publicly available.[2]

---

[2]https://github.com/SEU-COIN/K-DeCore

Table 1: Experimental results for comparison with baselines.

| Backbone | Method | SKR stream1 | | | SKR stream2 | | | SKR stream3 | | |
|---|---|---|---|---|---|---|---|---|---|---|
| | | AA | BWT | FWT | AA | BWT | FWT | AA | BWT | FWT |
| T5-LARGE | FINE-TUNING | 2.9 | −31.1 | 6.3 | 10.2 | −24.0 | 7.6 | 6.8 | −23.3 | 4.2 |
| | EMAR [19] | 12.6 | −18.5 | 4.8 | 12.5 | −19.5 | 7.2 | 8.7 | −20.6 | 3.0 |
| | O-LORA [11] | 1.4 | −30.5 | −2.3 | 7.8 | −17.9 | −5.4 | 5.4 | −27.8 | 0.3 |
| | PROGPROMPT[14] | 9.7 | −23.5 | 1.8 | 7.8 | −17.9 | 4.6 | 15.9 | −19.3 | 1.9 |
| | SFNET [9] | 27.3 | −14.7 | 3.6 | 18.2 | −27.5 | 4.2 | 24.3 | −18.0 | 3.1 |
| | C3 [12] | 26.6 | - | −1.9 | 30.8 | - | 4.2 | 29.8 | - | 3.2 |
| | SAPT [13] | 18.5 | −15.8 | 3.7 | 24.5 | −10.2 | 5.4 | 25.6 | −20.7 | 3.5 |
| | K-DECORE (Ours) | 31.8 | −9.6 | 8.6 | 31.4 | −7.5 | 8.4 | 30.1 | −6.9 | 7.7 |
| | MULTI-TASK | 37.4 | 9.1 | 8.8 | 38.4 | 12.6 | 10.1 | 36.6 | 11.5 | 6.8 |
| LLAMA3 -8B-INSTRUCT | FINE-TUNING | 22.8 | −29.4 | 4.2 | 26.3 | −22.9 | 1.8 | 16.4 | −29.1 | 1.6 |
| | EMAR [19] | 34.0 | −13.2 | 3.3 | 34.5 | −18.4 | 3.7 | 29.8 | −19.9 | 0.5 |
| | SFNET [9] | 36.1 | −14.5 | 3.7 | 35.8 | −17.1 | 4.2 | 35.4 | −12.3 | 4.2 |
| | C3 [12] | 39.7 | - | 2.3 | 40.7 | - | 2.6 | 36.6 | - | 2.1 |
| | K-DECORE (Ours) | 40.5 | −16.7 | 5.9 | 41.1 | −17.1 | 6.1 | 37.0 | −19.3 | 4.2 |
| | MULTI-TASK | 54.5 | 5.7 | 10.0 | 54.7 | 12.8 | 5.1 | 54.3 | 12.2 | 4.8 |
| QWEN2.5 -7B-INSTRUCT | FINE-TUNING | 19.6 | −26.9 | 5.1 | 28.8 | −15.9 | 4.6 | 25.3 | −22.8 | 1.9 |
| | EMAR [19] | 35.6 | −9.2 | 5.4 | 35.4 | −10.8 | 5.3 | 30.7 | −14.8 | 2.0 |
| | SFNET [9] | 40.3 | −10.5 | 4.5 | 38.5 | −13.2 | 4.8 | 34.1 | −15.0 | 5.1 |
| | C3 [12] | 38.9 | - | 1.9 | 38.8 | - | 1.8 | 35.9 | - | 2.6 |
| | K-DECORE (Ours) | 43.2 | −8.2 | 6.9 | 40.1 | −9.8 | 5.5 | 36.8 | −16.7 | 6.9 |
| | MULTI-TASK | 52.0 | 9.2 | 10.4 | 51.8 | 9.9 | 9.7 | 55.0 | 15.6 | 7.3 |

## 4.2 Overall Results

Table 1 presents a comprehensive performance analysis of our proposed K-DECORE framework against several baselines across three continual SKR streams, utilizing various LLM backbones. With T5 as the backbone, K-DECORE uniformly outperforms all competing methods across all evaluation metrics. This superior performance is largely maintained when employing Llama3 and QWEN2.5 as backbones, where K-DECORE establishes new state-of-the-art results in both AA and FWT.

In contrast, while PEFT-based methods such as C3 and SAPT achieve competitive results by dedicating independent parameters to each task, their per-task efficacy is hampered by the inherent convergence limitations of prompt-tuning. K-DECORE, however, attains superior performance without introducing additional parameters, offering a more efficient and scalable solution through its novel knowledge decoupling mechanism. Furthermore, our synthetic sample generation strategy, by creating more diverse query structures, enhances the model's generalization capabilities, as evidenced by its strong FWT performance. Notably, since C3 trains and loads a distinct PEFT module for each task, thereby precluding knowledge transfer to previous tasks, we do not report its BWT.

## 4.3 Ablation Study

To explore the contributions of each component of our proposed K-DECORE, we compared the performance of the following settings: **a) w/o Decoupling**: We omit the *knowledge decoupling* process and instead treat SKR as a standalone stage. In this configuration, we use a single model $f_\theta$ equipped with one PEFT module $\mathbf{P}$ for inference, while maintaining the replay process unchanged. **b) w/o Unification**: To evaluate the contribution of the *unified schema representation*, we only used the original schema format specific to each SKR task during the schema filtering stage. **c) w/o Replay**: We employ continual knowledge decoupling without replaying any samples, which involves removing the second term in both Equations (2) and (3). **d) w/o $\mathcal{M}_a^k$**: We removed the memory $\mathcal{M}_a^k$ in the continual schema filtering. **e) w/o $\mathcal{M}_b^k$**: We removed the memory $\mathcal{M}_b^k$ in the continual query building. **f) w Random Memory**: We randomly sample from $\mathcal{D}_a^k$ to construct $\mathcal{M}_a^k$ and $\mathcal{M}_b^k$.

Table 2 presents the results of our ablation study, investigating the contribution of key components of K-DECORE. Removing the entire replay mechanism or specifically the query structure memory (**w/o**

Table 2: Experimental results of ablation studies.

| Backbone | Method | SKR stream1 | | | SKR stream2 | | | SKR stream3 | | |
|---|---|---|---|---|---|---|---|---|---|---|
| | | AA | BWT | FWT | AA | BWT | FWT | AA | BWT | FWT |
| LLAMA3 -8B-INSTRUCT | K-DECORE | 40.5 | −16.7 | 5.9 | 41.1 | −17.1 | 6.1 | 37.0 | −19.3 | 4.2 |
| | w/o Decoupling | 38.8 | −13.8 | 2.6 | 37.5 | −19.8 | 2.7 | 33.6 | −19.6 | 2.1 |
| | w/o Unification | 37.8 | −17.7 | 3.4 | 37.1 | −15.2 | 0.7 | 34.6 | −15.7 | −0.8 |
| | w/o Replay | 20.5 | −41.2 | 4.3 | 28.8 | −34.4 | 6.8 | 17.9 | −43.8 | 3.5 |
| | w/o $\mathcal{M}_a^k$ | 39.4 | −17.1 | 5.1 | 38.2 | −20.9 | 6.2 | 35.6 | −20.2 | 3.6 |
| | w/o $\mathcal{M}_b^k$ | 20.8 | −42.1 | 5.3 | 28.6 | −32.8 | 5.3 | 18.5 | −44.5 | 3.7 |
| | w Random Memory | 39.9 | −17.4 | 5.6 | 37.7 | −20.6 | 5.4 | 36.3 | −21.3 | 3.9 |
| QWEN2.5 -7B-INSTRUCT | K-DECORE | 43.2 | −8.2 | 6.9 | 40.1 | −9.8 | 5.5 | 36.8 | −16.7 | 6.9 |
| | w/o Decoupling | 37.9 | −8.8 | 5.4 | 39.3 | −5.5 | 4.0 | 36.1 | −17.1 | 4.8 |
| | w/o Unification | 34.4 | −13.3 | 1.8 | 37.4 | −10.5 | 2.6 | 36.1 | −11.7 | 2.4 |
| | w/o Replay | 16.8 | −41.4 | 5.3 | 29.7 | −23.8 | 5.5 | 18.7 | −40.3 | 6.5 |
| | w/o $\mathcal{M}_a^k$ | 42.8 | −9.7 | 5.3 | 39.5 | −9.8 | 4.9 | 35.9 | −17.5 | 6.6 |
| | w/o $\mathcal{M}_b^k$ | 19.1 | −39.2 | 6.0 | 28.3 | −24.4 | 4.7 | 16.3 | −43.8 | 6.8 |
| | w Random Memory | 42.1 | −7.7 | 5.3 | 39.4 | −11.8 | 6.3 | 36.2 | −17.9 | 5.7 |

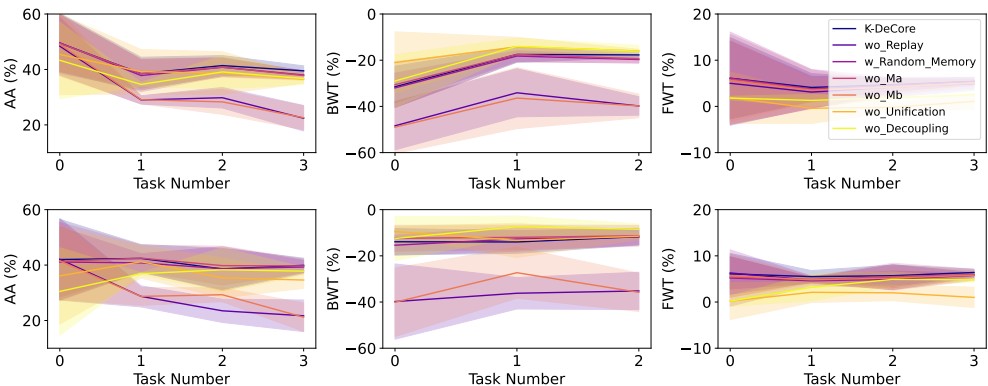

Figure 3: AA (%), BWT (%), and FWT (%) till the seen tasks after learning on each task, using LLAMA3-8B (top row) and QWEN2.5-8B (bottom row). Solid lines represent the mean values across three distinct task sequences, while shaded regions indicate the standard deviation.

$\mathcal{M}_b^k$) leads to drastic performance drops in AA and significantly worse BWT, highlighting the critical role of memory, particularly the storage of diverse query structures, in mitigating forgetting and maintaining overall performance. Disabling the knowledge decoupling (**w/o Decoupling**) or removing the unified schema representation (**w/o Unification**) also results in lower AA and reduced FWT, demonstrating that separating reasoning into distinct stages and standardizing schema representation are vital for effective knowledge transfer and handling heterogeneous tasks. While removing the schema memory (**w/o $\mathcal{M}_a^k$**) or using random memory sampling shows less severe degradation compared to removing query memory, performance still drops relative to the full model, confirming the benefits of dedicated schema memory and our proposed memory construction strategy.

### 4.4 Performance Till the Seen Tasks

Figure 3 presents the performance trajectory of the evaluated methods on previously seen tasks across different LLM backbones. Our proposed K-DECORE framework consistently demonstrates superior overall performance, with its advantage becoming more pronounced as the number of sequential tasks increases. The model's robust forward transfer capabilities, a key contributor to this success, can be attributed to our strategies for pseudo-sample synthesis and continuous query construction, which collectively enhance its adaptability to new and unseen scenarios. An interesting trade-off is revealed in our ablation study. While omitting the knowledge decoupling component results in a higher BWT, this apparent benefit comes at the cost of increased inter-task interference. This interference ultimately diminishes the model's adaptability, leading to poorer performance in FWT and overall AA, thereby underscoring the critical role of our decoupling strategy.

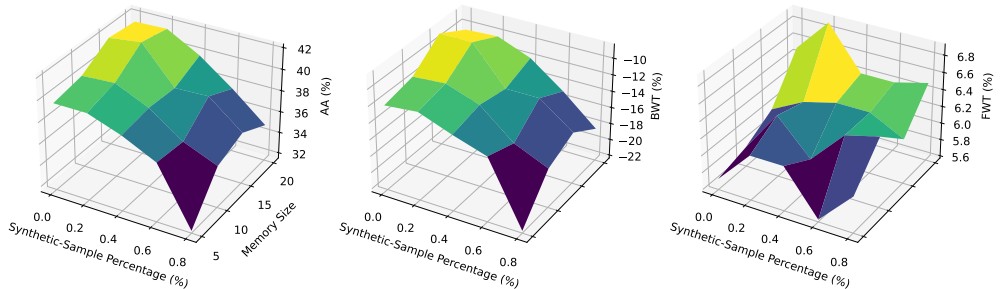

Figure 4: Peformance of K-DECORE with varing memory sizes and synthetic sample percentages.

Table 3: Experimental results on imbalanced task streams.

| Backbone | Method | SKR stream1 | | | SKR stream2 | | | SKR stream3 | | |
|---|---|---|---|---|---|---|---|---|---|---|
| | | AA | BWT | FWT | AA | BWT | FWT | AA | BWT | FWT |
| QWEN2.5 -7B-INSTRUCT | FINE-TUNING | 18.5 | −15.2 | 4.1 | 22.2 | −11.8 | 4.8 | 26.3 | −5.1 | 3.4 |
| | EMAR [19] | 28.2 | −3.1 | 5.4 | 30.7 | −1.1 | 5.1 | 28.3 | −1.6 | 3.2 |
| | K-DECORE (Ours) | 31.8 | −10.9 | 8.5 | 33.5 | −8.5 | 7.3 | 30.1 | −6.5 | 6.2 |

## 4.5 Effects of Varying Memory Sizes and Synthetic Sample Percentages

Figure 4 shows the AA, BWT, and FWT of various methods across seen tasks using different backbone LLMs. When the proportion of pseudo samples is relatively low, increasing the memory size tends to enhance AA and BWT. Conversely, with a fixed memory size, incorporating pseudo samples at a proportion of 20% frequently yields optimal results in terms of AA, BWT, and FWT, outperforming scenarios with either no pseudo samples or a higher proportion of them. Interestingly, for FWT, a larger memory capacity does not inherently translate to improved generalization capabilities.

## 4.6 Performance on Imbalanced Task Streams

To further assess our model's robustness in more realistic scenarios, we conducted an additional experiment simulating an imbalanced data stream, a common challenge where tasks have a non-uniform number of training samples. Specifically, we configured the training set with the following sample distribution: Spider (500), GrailQA (800), CombWebQ (300), and MTOP (600). Here we use QWEN2.5-7B-INSTRUCT as the backbone model. As shown in Table 3, our method, KE-DECORE, demonstrates clear superiority in the imbalanced data stream experiment. It achieves the highest AA on all the streams, significantly outperforming the EMAR baseline.

## 4.7 Training Efficiency

Figure 5 illustrates the average training duration for each approach across various tasks. Notably, C3 exhibited a considerably longer training time compared to other methods, primarily due to its prompt tuning necessitating an extensive number of epochs. The primary source of

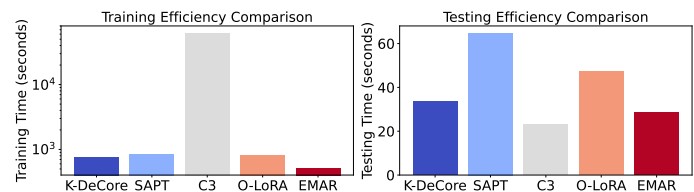

Figure 5: Training and testing time of various methods.

time overhead for SAPT and O-LoRA lies in the merging process of multiple LoRA modules. This bottleneck becomes increasingly pronounced as the number of tasks scales up. In contrast, our K-DECORE demonstrated a favorable balance between efficiency and performance, with its training time only marginally exceeding that of the rehearsal-based EMAR.

# 5  Related Work

## 5.1  Structured Knowledege Reasoning.

Structured Knowledge-intensive Retrieval (SKR) tasks, which include text-to-SQL [21], Knowledge Graph Question Answering (KGQA), and dialogue parsing, are centered on converting natural language questions (NLQs) into structured, executable queries. In the database domain (DB SKR), the process of translating NLQs into SQL queries has advanced significantly. Initial methods relied on specialized model architectures [22] and the use of intermediate representations [23]. More recently, the field has shifted towards leveraging Large Language Models (LLMs) enhanced with sophisticated techniques such as task decomposition, chain-of-thought prompting [24, 25], and self-consistency [26]. This modern approach has led to substantial performance gains, as demonstrated in several recent studies [27, 28, 29, 30]. Similarly, SKR over knowledge graphs (KG SKR), or KGQA, has evolved from traditional semantic parsing [31, 32] and embedding-based methods [33]. Recent advancements, such as DecAF [34] and KB-BINDER [35], now integrate the generation of logical forms with direct answer retrieval, markedly improving performance. In the domain of dialogue parsing, tasks like TOP [36] and MTOP [18] involve deciphering complex, multi-turn interactions. These have also been advanced by LLMs, which provide a more nuanced understanding of the required structured representations.

While APEX [10] and LECSP [37] also constitute related work, they were excluded from our comparative analysis. These methods are specifically tailored for the text-to-SQL task and do not generalize to the other SKR domains we address. Furthermore, their computational overhead when implemented with LLM backbones exceeds the practical limits of our experimental setup.

## 5.2  LLM Continual Learning.

In the context of continual learning for LLMs [38], Parameter-Efficient Fine-Tuning (PEFT) methods, including prompt tuning [39], adapters [40], and LoRA [20], have become instrumental. These techniques are particularly effective for adapting models to new tasks with limited data [41, 10] by introducing a small number of trainable parameters while keeping the base model frozen. This parameter-efficient approach allows for the sequential integration of new knowledge, mitigating catastrophic forgetting and adapting to distribution shifts without the need for replaying historical data [12, 13]. A common strategy in continual learning is to train a separate PEFT module for each new task. This compartmentalizes task-specific knowledge, effectively preventing interference between tasks [42, 13, 43].

However, this one-to-one mapping of modules to tasks presents a significant drawback: it inherently struggles with generalization to novel samples that do not neatly fit into the learned task distributions, thereby limiting its practical utility in dynamic, real-world scenarios. In stark contrast, our proposed K-DECORE framework utilizes a fixed set of PEFT modules throughout the entire learning process. This design offers a more efficient and scalable solution for both training and inference in evolving, iterative environments.

# 6  Conclusion

In this paper, we introduced K-DECORE, a novel framework for Continual Structured Knowledge Reasoning (CSKR) that effectively addresses the challenges of catastrophic forgetting and parameter growth in heterogeneous structured knowledge environments. By leveraging knowledge decoupling, K-DECORE decouples schema filtering from query construction, enabling efficient knowledge transfer across diverse tasks without increasing model parameters. Our dual-perspective memory mechanism further enhances knowledge retention by maintaining replay samples from both schema and query structure perspectives, while our query synthesis strategy enriches memory content with complex structured queries. Through extensive evaluations on task streams from four SKR benchmarks, K-DECORE consistently demonstrated superior performance over strong baselines across multiple metrics. However, due to time and cost constraints, we did not employ reasoning-based LLMs (like QWEN3) as the backbone for our framework. This remains a limitation of our current work. Future research will explore the integration of such models to potentially enhance the applicability and robustness of K-DECORE in real-world scenarios.

# 7 Acknowledgements

This work was partially funded by National Nature Science Foundation of China (Grant No. U21A20488, No. 62376058, No. 62476058, and No. 6250072425), and funded by Southeast University-China Mobile Research Institute Joint Innovation Center and the Southeast University Interdisciplinary Research Program for Young Scholars. We thank the Big Data Computing Center of Southeast University for providing the facility support on the numerical calculations in this paper.

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

# A  DB-style Input and Output Format

We emphasize that our DB-style unification is designed to fully preserve all critical information, ensuring that the flattened representation remains faithful to the original structures. Below, we explain how our design achieves this preservation while enabling effective covering across tasks.

For relation direction, our mapping explicitly preserves the subject-object orientation inherent in triples (subject-predicate-object). Specifically, we designate the content in the primary key of the transformed table as the subject, while the contents in other columns represent the corresponding objects. This structured encoding ensures that directional information is retained and can be reliably interpreted during query generation, preventing ambiguities that might arise from undirected representations.

Regarding edge types, we incorporate them directly into the schema by prefixing each relation (or edge) with its type and treating it as a dedicated column in the flattened table. This simple yet effective augmentation embeds the type information within the schema itself, allowing the model to access and utilize it without requiring additional modifications to the backbone architecture.

For multi-level structures, such as hierarchical slots or nested tables, we adopt flattening techniques. This involves recursively expanding multi-level columns into single-level ones, ensuring that hierarchical relationships are linearized but not discarded. Our framework is flexible enough to accommodate this, as the flattening occurs preprocessing and does not alter the core continual learning mechanism.

More crucially, our DB-style unification is applied only during the schema filtering stage, where the primary objective is akin to entity linking—namely, identifying the relevant tables (entity types) and columns (relations) for the given query. In this context, not all granular details (e.g., full hierarchical depth) are necessary, as the filter's role is to select pertinent schema elements rather than reconstruct the entire original structure. The subsequent query builder module, operating on the filtered schema, can then draw on the preserved essential information.

## A.1  GrailQA

**Schema Filtering Stage:**

```
INPUT:
Identify the exact schema component from the given schema that would correctly answer the
↪  following question.

schema:

book.series_editor : book_edition_series_edited | book.book_edition : book_edition_series,
↪  place_of_publication | book.book_edition_series : editions_in_this_series, series_editor

question:

a people's history of christianity was edited by what series editor?

OUTPUT:
book.series_editor : book_edition_series_edited
```

**Query Building Stage:**

```
INPUT:
Generate an s-expression that can be used to find the answer to the following question using the
↪  knowledge graph schema items provided.

schema:

a people's history of christianity: m.012bphrj | book.series_editor : book_edition_series_edited |
↪  book.book_edition : book_edition_series, place_of_publication | book.book_edition_series :
↪  editions_in_this_series, series_editor

question:

a people's history of christianity was edited by what series editor?

OUTPUT:
(AND book.series_editor (JOIN book.series_editor.book_edition_series_edited m.012bphrj))
```

## A.2   Spider

**Schema Filtering Stage:**

```
INPUT:
Identify the exact schema component from the given schema that would correctly answer the
↪  following question.

schema:

department : department_id , name , creation , ranking , budget_in_billions , num_employees | head
↪  : head_id , name , born_state , age | management : department_id , head_id , temporary_acting

question:

How many heads of the departments are older than 56 ?

OUTPUT:
head : age
```

**Query Building Stage:**

```
INPUT:
Your task is to convert the following question to an SQL query using the following database
↪  schema.

schema:
department : department_id , name , creation , ranking , budget_in_billions , num_employees | head
↪  : head_id , name , born_state , age | management : department_id , head_id , temporary_acting

question:
How many heads of the departments are older than 56 ?

OUTPUT:
select count(*) from head where age > 56
```

## A.3   CWQ

**Schema Filtering Stage:**

```
INPUT:
Identify the exact schema component from the given schema that would correctly answer the
↪  following question.

schema:

type.object : type | location.location : time_zones, containedby, contains, people_born_here |
↪  common.topic : notable_types | geography.river : mouth | travel.tourist_attraction :
↪  near_travel_destination | kg.object_profile : prominent_type |
↪  education.educational_institution : colors | location.country : languages_spoken,
↪  administrative_divisions

question:What language is spoken in the country that has Southern Peninsular?

OUTPUT:
location.country : languages_spoken, administrative_divisions
```

**Query Building Stage:**

```
INPUT:
Given the list of schema items, write an SPARQL query that can be used to find the answer to the
↪  following question.

schema:
Southern Peninsula: m.08kmfj | type.object : type | location.location : time_zones, containedby,
↪  contains, people_born_here | common.topic : notable_types | geography.river : mouth |
↪  travel.tourist_attraction : near_travel_destination | kg.object_profile : prominent_type |
↪  education.educational_institution : colors | location.country : languages_spoken,
↪  administrative_divisions

question:
What language is spoken in the country that has Southern Peninsular?

OUTPUT:
PREFIX ns: <http://rdf.freebase.com/ns/>SELECT DISTINCT ?xWHERE {?c
↪  ns:location.country.administrative_divisions ns:m.08kmfj .?c
↪  ns:location.country.languages_spoken ?x .}
```

## A.4 MTOP

**Schema Filtering Stage:**

```
INPUT:

Identify the exact schema component from the given schema that would correctly answer the
↪  following question.

schema:
in:get : message, weather, alarm, info_recipes, stories_news, reminder, recipes, event, call_time,
↪  life_event, info_contact, contact, timer, reminder_date_time, age, sunrise, employer,
↪  education_time, job, availability, category_event, call, employment_time, call_contact,
↪  location, track_info_music, sunset, mutual_friends, undergrad, reminder_location,
↪  attendee_event, message_contact, reminder_amount, date_time_event, details_news,
↪  education_degree, major, contact_method, life_event_time, lyrics_music, airquality, language,
↪  gender, group | in:send_message | in:set : unavailable, rsvp_yes, available,
↪  default_provider_music, rsvp_interested, default_provider_calling, rsvp_no | in:delete :
↪  reminder, alarm, timer, playlist_music | in:create : alarm, reminder, call, playlist_music,
↪  timer | in:question : news, music | in:play : music, media | in:end_call | in:ignore_call |
↪  in:update_call | in:update_reminder_date_time | in:pause : music, timer | in:answer_call |
↪  in:snooze_alarm | in:update_reminder_todo | in:is_true_recipes | in:remove_from_playlist_music
↪  | in:add : time_timer, to_playlist_music | in:share_event | in:prefer | in:start_shuffle_music
↪  | in:silence_alarm | in:switch_call | in:subtract_time_timer | in:update_timer |
↪  in:previous_track_music | in:hold_call | in:skip_track_music | in:update_method_call |
↪  in:update_alarm | in:like_music | in:restart_timer | in:resume : timer, call, music |
↪  in:merge_call | in:replay_music | in:loop_music | in:stop : music, shuffle_music |
↪  in:unloop_music | in:update_reminder_location | in:cancel : message, call | in:update_reminder
↪  | in:rewind_music | in:repeat : all_music, all_off_music | in:fast_forward_music |
↪  in:dislike_music | in:disprefer | in:help_reminder | in:follow_music

question:
Has Angelika Kratzer video messaged me ?

OUTPUT:

IN:GET : MESSAGE
```

## Query Building Stage:

```
INPUT:
Convert the following natural language query to an API call in Task Oriented Parsing (TOP)
↪  representation using the following API specification.

API specification:
IN:GET : MESSAGE, WEATHER, ALARM, INFO_RECIPES, STORIES_NEWS, REMINDER, RECIPES, EVENT, CALL_TIME,
↪  LIFE_EVENT, INFO_CONTACT, CONTACT, TIMER, REMINDER_DATE_TIME, AGE, SUNRISE, EMPLOYER,
↪  EDUCATION_TIME, JOB, AVAILABILITY, CATEGORY_EVENT, CALL, EMPLOYMENT_TIME, CALL_CONTACT,
↪  LOCATION, TRACK_INFO_MUSIC, SUNSET, MUTUAL_FRIENDS, UNDERGRAD, REMINDER_LOCATION,
↪  ATTENDEE_EVENT, MESSAGE_CONTACT, REMINDER_AMOUNT, DATE_TIME_EVENT, DETAILS_NEWS,
↪  EDUCATION_DEGREE, MAJOR, CONTACT_METHOD, LIFE_EVENT_TIME, LYRICS_MUSIC, AIRQUALITY, LANGUAGE,
↪  GENDER, GROUP | IN:SEND_MESSAGE | IN:SET : UNAVAILABLE, RSVP_YES, AVAILABLE,
↪  DEFAULT_PROVIDER_MUSIC, RSVP_INTERESTED, DEFAULT_PROVIDER_CALLING, RSVP_NO | IN:DELETE :
↪  REMINDER, ALARM, TIMER, PLAYLIST_MUSIC | IN:CREATE : ALARM, REMINDER, CALL, PLAYLIST_MUSIC,
↪  TIMER | IN:QUESTION : NEWS, MUSIC | IN:PLAY : MUSIC, MEDIA | IN:END_CALL | IN:IGNORE_CALL |
↪  IN:UPDATE_CALL | IN:UPDATE_REMINDER_DATE_TIME | IN:PAUSE : MUSIC, TIMER | IN:ANSWER_CALL |
↪  IN:SNOOZE_ALARM | IN:UPDATE_REMINDER_TODO | IN:IS_TRUE_RECIPES | IN:REMOVE_FROM_PLAYLIST_MUSIC
↪  | IN:ADD : TIME_TIMER, TO_PLAYLIST_MUSIC | IN:SHARE_EVENT | IN:PREFER | IN:START_SHUFFLE_MUSIC
↪  | IN:SILENCE_ALARM | IN:SWITCH_CALL | IN:SUBTRACT_TIME_TIMER | IN:UPDATE_TIMER |
↪  IN:PREVIOUS_TRACK_MUSIC | IN:HOLD_CALL | IN:SKIP_TRACK_MUSIC | IN:UPDATE_METHOD_CALL |
↪  IN:UPDATE_ALARM | IN:LIKE_MUSIC | IN:RESTART_TIMER | IN:RESUME : TIMER, CALL, MUSIC |
↪  IN:MERGE_CALL | IN:REPLAY_MUSIC | IN:LOOP_MUSIC | IN:STOP : MUSIC, SHUFFLE_MUSIC |
↪  IN:UNLOOP_MUSIC | IN:UPDATE_REMINDER_LOCATION | IN:CANCEL : MESSAGE, CALL | IN:UPDATE_REMINDER
↪  | IN:REWIND_MUSIC | IN:REPEAT : ALL_MUSIC, ALL_OFF_MUSIC | IN:FAST_FORWARD_MUSIC |
↪  IN:DISLIKE_MUSIC | IN:DISPREFER | IN:HELP_REMINDER | IN:FOLLOW_MUSIC

question:
Has Angelika Kratzer video messaged me ?

OUTPUT:

[IN:GET_MESSAGE [SL:CONTACT Angelika Kratzer ] [SL:TYPE_CONTENT video ] [SL:RECIPIENT me ] ]
```

# B Prompts for Synthesizing Query Structures

## B.1 Prompt for Pseudo Structure Synthesis

```
You are an expert in logical query expression synthesis. Your task is to merge two simple query
↪  skeletons into a single, more complex skeleton that logically integrates their structure and
↪  meaning. The result should preserve the semantics of both inputs and follow the same
↪  structural style and syntax. Output only the composed skeleton. Do not include any explanation
↪  or additional text.

Simple skeleton 1:\{example1\}

Simple skeleton 2:\{example2\}

Composed skeleton:
```

## B.2 Prompt for Pseudo Question Generation

```
You are an AI assistant that specializes in transforming structured query statements into fluent,
↪  contextually accurate natural language questions. Your task is to read a given formal query
↪  and its associated schema context, and then write a corresponding question in clear, natural
↪  language that would retrieve the correct answer if the formal query were executed on a
↪  database.

question:
{query}

schema:
{schema}

Follow these steps carefully:

1. Understand the Query Semantics:
Analyze the query logic, including the target variables, the filtering conditions, the joins, and
↪  the structure of the query. Infer what specific information the query is intended to retrieve.

2. Incorporate Schema Semantics:
Use the provided schema elements (such as table names, entity types, or relation labels) to guide
↪  the terminology and phrasing in the question. Map schema components to human-interpretable
↪  phrases as necessary.

3. Avoid Direct Translation:
Do not simply convert the query structure into a rigid, mechanical form. Instead, aim for a fluent,
↪  natural-sounding question that a human might ask when seeking the same information.

4. Preserve Answer Equivalence:
Ensure that the generated question is logically equivalent to the formal query in terms of the
↪  expected answer. The question should not broaden, narrow, or alter the scope of the query.

5. Maintain Clarity and Brevity:
The question should be unambiguous and concise, while preserving all the critical information
↪  needed to align with the query.

6. Output Format Constraint:
Only output the final question, without any explanatory text, metadata, or formatting symbols. The
↪  output should consist solely of a single fluent question in English.
```

## B.3 Examples of Structure Synthesis

```
Structure 1:
(ARGMAX [T1] [C1])

Structure 2:
(AND [T1] (JOIN [C1] [E1]))

Synthetic Structure:
(ARGMAX (AND [T1] (JOIN [C1] [E1])) [C2])
```

```
Structure 1:
SELECT [C1] FROM [T1] WHERE [C2] = [V1];

Structure 2:
SELECT [C1] FROM [T] WHERE [C3] IN [V1];

Synthetic Structure:
SELECT * FROM [T1] WHERE [C1] IN (SELECT [C2] FROM [T2] WHERE [C3] = [V1]);
```

## C  Hyperparameter settings

Our hyperparameter settings are as follows:

Table 4: Hyperparameter Configuration for Experiments

| Hyperparameter | Value |
| --- | --- |
| GPU | NVIDIA RTX 4090 |
| PEFT Module | LoRA [20] |
| LoRA Target | gate_proj, down_proj, up_proj, q_proj, v_proj, k_proj, o_proj |
| LoRA Rank | 8 |
| LoRA $\alpha$ | 16 |
| Memory Size per Task | $|\mathcal{M}_a^k| = 5, |\mathcal{M}_b^k| = 5$ |
| Real to Pseudo Memory Ratio | $4:1$ |
| Batch Size | 12 |
| Global Batch Size | 32 |
| Learning Rate | $5 \times 10^{-5}$ |
| Optimizer | Adam |
| Training Epochs | 5 |
| Maximum Input Length | 1024 |
| Maximum Output Length | 256 |
| Parameter $T$ in Algorithm 1 | 5 |

