# OpenReview forum: "K-DeCore: Facilitating Knowledge Transfer in Continual Structured Knowledge Reasoning via Knowledge Decoupling"
_NeurIPS.cc/2025/Conference — NeurIPS 2025 poster_

### Official Review · Reviewer_Hfgq · 2025-06-25

**Clarity:** 3
**Significance:** 2
**Originality:** 2
**Rating:** 4
**Confidence:** 2

**Summary:**

The work studies continual structured knowledge reasoning, where a model must translate natural‑language questions into structured queries while it moves through a sequence of tasks that involve different forms of knowledge, such as relational tables, knowledge graphs, and dialogue states. Current continual learning methods either add new parameters for each task or generalize poorly across heterogeneous schemas. The authors introduce K‑DeCore, which keeps one frozen backbone language model and attaches two small, trainable modules: a schema filter that selects the relevant parts of a given schema and a query builder that forms the final query from the filtered schema. By separating these two stages, the approach transfers knowledge across tasks without increasing the parameter budget. K‑DeCore also maintains dual replay memories, one that stores representative schema examples and another that stores diverse query structures, and enriches them with synthetically generated queries to raise coverage.

**Questions:**

Please refer to the Strengths And Weaknesses.

**Ethical Concerns:**

["NO or VERY MINOR ethics concerns only"]

**Final Justification:**

My concerns have been addressed after reading the response. I would like to maintain my positive recommendation for this paper.

**Limitations:**

Yes.

**Paper Formatting Concerns:**

No.

**Quality:**

2

**Strengths And Weaknesses:**

**Strengths**

- The paper frames continual structured knowledge reasoning as a sequence of heterogeneous tasks and tackles it with a fixed‑parameter backbone plus two small LoRA modules, one for schema filtering and one for query building. By separating these roles, the design allows the same parameters to serve databases, knowledge graphs, and dialogue states without growing with the number of tasks, directly addressing the scalability problem that affects many prior rehearsal and adapter methods.

- The knowledge decoupling idea is paired with a dual‑perspective replay memory: schema‑guided clusters preserve coverage of previously seen schema elements, while structure‑guided clusters store diverse query skeletons. This two‑view memory gives a clear, task‑specific reason for each stored sample and is lightweight, which keeps continual training practical on a single GPU.

- Experiments span four public datasets, formed into three task streams, and the study tests three backbones.

**Weaknesses**

- Both schema‑guided and structure‑guided memories use cosine distance over embeddings from a separate encoder, which may leave important but embedding‑close cases unprotected.

- Mapping every schema into a flat table format removes relation direction, edge types, and slot hierarchy.

- The query-building generator stitches together random structure templates and schema fragments and then discards any sample that fails to execute. This generate-and-filter loop can bias memory toward structures that are easier for the SQL engine to validate rather than those that improve model generalization.

---

> ### Author Rebuttal · Authors · 2025-07-30
>
> We are very grateful to you for providing us with valuable feedback and suggestions for our paper. We will provide explanations and clarifications for each weakness and question.
>
>
> ### **Weakness 1:**
> Both schema‑guided and structure‑guided memories use cosine distance over embeddings from a separate encoder, which may leave important but embedding‑close cases unprotected.
>
>
> ### **Response for Weakness 1:**
> The primary goal of employing cosine distance in both the schema-guided and structure-guided memories is to identify and prioritize samples that exhibit significant differences in their embedding space, thereby capturing a broad representation of the dataset’s overall distribution. This approach is grounded in the assumption that, **for samples with highly similar embedding semantics (i.e., those that are close in the embedding space), retaining just one representative instance is often sufficient for effective continual learning.** By focusing on diversity through distance maximization, we aim to select the most comprehensive set of exemplars within the constraints of a limited memory budget. This not only ensures coverage of common patterns but also, in theory, allows our method to emphasize long-tail samples—those rarer or more unique cases that might otherwise be overlooked in denser regions of the embedding space.
>
> ### **Weakness 2:**
>
> Mapping every schema into a flat table format removes relation direction, edge types, and slot hierarchy.
>
>
> ### **Response for Weakness 2:**
>
> We emphasize that our DB-style unification is designed to fully preserve all critical information, ensuring that the flattened representation remains faithful to the original structures. Below, we explain how our design achieves this preservation while enabling effective covering across heterogeneous tasks.
>
> For relation direction, our mapping explicitly preserves the subject-object orientation inherent in triples (subject-predicate-object). Specifically, we designate the content in the primary key of the transformed table as the subject, while the contents in other columns represent the corresponding objects. This structured encoding ensures that directional information is retained and can be reliably interpreted during query generation, preventing ambiguities that might arise from undirected representations.
>
> Regarding edge types, we incorporate them directly into the schema by prefixing each relation (or edge) with its type and treating it as a dedicated column in the flattened table. This simple yet effective augmentation embeds the type information within the schema itself, allowing the model to access and utilize it without requiring additional modifications to the backbone architecture.
>
> For multi-level structures, such as hierarchical slots or nested tables, we adopt flattening techniques inspired by related works[1,2]. This involves recursively expanding multi-level columns into single-level ones, ensuring that hierarchical relationships are linearized but not discarded. Our framework is flexible enough to accommodate this, as the flattening occurs preprocessing and does not alter the core continual learning mechanism.
>
> More crucially, our DB-style unification is applied **only during the schema filtering stage**, where the primary objective is akin to entity linking—namely, identifying the relevant tables (entity types) and columns (relations) for the given query. In this context, not all granular details (e.g., full hierarchical depth) are necessary, as the filter's role is to select pertinent schema elements rather than reconstruct the entire original structure. The subsequent query builder module, operating on the filtered schema, can then leverage the frozen backbone to generate accurate queries, drawing on the preserved essential information.
>
> Our experimental results in Table 1 validate the robustness of our K-DeCore, demonstrating strong performance on the task streams over mainstream and widespread heterogeneous knowledge.
>
> We will explain in detail the transformation process of each structured knowledge in a camera-ready version once the paper is accepted.
>
> - [1] Min, Dehai, et al. "Exploring the Impact of Table-to-Text Methods on Augmenting LLM-based Question Answering with Domain Hybrid Data." Proceedings of the 2024 Conference of the North American Chapter of the Association for Computational Linguistics: Human Language Technologies (Volume 6: Industry Track). 2024.
> - [2] Zhang, Zhehao, Yan Gao, and Jian-Guang Lou. "e5: Zero-shot hierarchical table analysis using augmented LLMs via explain, extract, execute, exhibit and extrapolate." Proceedings of the 2024 Conference of the North American Chapter of the Association for Computational Linguistics: Human Language Technologies (Volume 1: Long Papers). 2024.
>
>
> ### **Weakness 3:**
>
> The query-building generator stitches together random structure templates and schema fragments and then discards any sample that fails to execute. This generate-and-filter loop can bias memory toward structures that are easier for the SQL engine to validate rather than those that improve model generalization.
>
>
> ### **Response for Weakness 3:**
>
> The core rationale for incorporating successful execution as a filtering criterion is to guarantee that each pseudo-sample added to the memory is semantically and syntactically valid. We firmly believe that including incorrect or invalid samples would introduce harmful noise into the replay buffer, potentially degrading the model's ability to learn robust patterns across tasks. By filtering out failures, we prioritize high-quality augmentations that faithfully represent plausible query-schema interactions, thereby supporting effective knowledge transfer without polluting the training data. Importantly, this process does not inherently favor "easier" structures; instead, it enforces a baseline of executability, which aligns with real-world query requirements and encourages the model to generalize to diverse, valid scenarios rather than overfitting to artificial or erroneous ones.
>
> Furthermore, we clarify that our approach is not limited to SQL engine execution. We adapt the validation to the specific nature of each dataset: for queries on the CWQ dataset, we utilize a SPARQL engine to execute and verify SPARQL queries; for the GrailQA and MTOP datasets, we leverage the official evaluation code provided by the original papers to assess query legality, ensuring a tailored and accurate check. This multi-engine strategy broadens the scope of validated structures, reducing any dataset-specific biases and promoting a more comprehensive memory that captures heterogeneous query forms.
>
> We will explain this section in detail in the camera-ready version once the paper is accepted.

---

> > ### Comment · Reviewer_Hfgq · 2025-08-05
> >
> > I thank the authors for the detailed rebuttal. My concerns have been addressed after reading the response. I would like to keep my score.

---

### Official Review · Reviewer_fB8B · 2025-07-01

**Clarity:** 4
**Significance:** 4
**Originality:** 3
**Rating:** 5
**Confidence:** 5

**Summary:**

K-DECORE addresses challenges in Continual Structured Knowledge Reasoning by decoupling reasoning into task-specific (query building) and task-agnostic (schema filtering) components. It uses a fixed parameter model and integrates memory consolidation and pseudo-data synthesis for better generalization and reduced catastrophic forgetting. Experiments on benchmark datasets with LLM backbones show K-DECORE’s superiority over existing methods.

**Questions:**

Could you elaborate on why the "limited convergence of prompt-tuning" (for methods like C3 and SAPT) poses a constraint, and how K-DECORE's decoupled design inherently overcomes this issue, leading to its superior results without merely introducing additional parameters?

**Ethical Concerns:**

["NO or VERY MINOR ethics concerns only"]

**Final Justification:**

The authors have addressed most of my concerns and I would like to keep my score.

**Limitations:**

Yes.

**Quality:**

3

**Strengths And Weaknesses:**

Strength:

1. The paper makes a significant contribution by being the first to systematically explore continual learning across heterogeneous Structured Knowledge Reasoning tasks. This addresses a crucial real-world challenge where models need to continuously adapt to new reasoning tasks with varying structured knowledge types.

2. The core idea of decoupling the reasoning process into schema filtering (task-agnostic) and query building (task-specific) is highly innovative and well-motivated. This design effectively bridges gaps across diverse tasks by identifying schema filtering as a shared, reusable component.

3.  K-DECORE operates with a fixed number of tunable parameters, which is a major advantage over prior methods that suffer from parameter growth as tasks increase. This contributes to more efficient reasoning and scalability.

4. The experimental results are comprehensive and convincingly demonstrate K-DECORE's superior performance across all evaluation metrics (AA, BWT, FWT) compared to strong baselines, including rehearsal-based and PEFT-based methods. The use of three different LLM backbones further strengthens the validity of the findings.

Weakness:
1. While K-DECORE significantly improves BWT compared to baselines like FINE-TUNING, the BWT scores for LLAMA3 and QWEN2.5 backbones are still negative (e.g., -16.7 for Llama3 Stream1, -8.2 for Qwen2.5 Stream1). This indicates that catastrophic forgetting, while mitigated, remains a persistent challenge. The BWT sometimes remains comparable to or slightly worse than some rehearsal-based methods (e.g., EMAR) in certain streams for Llama3.

2. For the structure-guided query synthesis, the paper mentions that novel structures are synthesized "guided by carefully curated demonstrations" . While Appendix B is referenced for examples, more explicit details on demonstration would enhance reproducibility and understanding of this crucial step.

---

> ### Author Rebuttal · Authors · 2025-07-30
>
> We are very grateful to you for providing us with valuable feedback and suggestions for our paper. We will provide explanations and clarifications for each weakness and question.
>
>
> ### **Weakness 1:**
> While K-DECORE significantly improves BWT compared to baselines like FINE-TUNING, the BWT scores for LLAMA3 and QWEN2.5 backbones are still negative (e.g., -16.7 for Llama3 Stream1, -8.2 for Qwen2.5 Stream1). This indicates that catastrophic forgetting, while mitigated, remains a persistent challenge. The BWT sometimes remains comparable to or slightly worse than some rehearsal-based methods (e.g., EMAR) in certain streams for Llama3.
>
> ### **Response for Weakness 1:**
> First, it's important to emphasize that K-DeCore delivers significant improvements in BWT across all evaluated backbones compared to the Fine-Tuning baseline, regardless of the model (e.g., Llama3, Qwen2.5, or others). This consistent enhancement demonstrates that catastrophic forgetting is effectively alleviated, with our decoupled schema filtering and query building modules, combined with dual replay memories, enabling better retention of prior knowledge without parameter expansion. Moreover, when assessing the overall performance via the Average Accuracy (AA) metric—which captures the holistic effect across the entire task stream—K-DeCore outperforms **all compared methods**, including rehearsal-based approaches like EMAR. This suggests that while isolated BWT metrics may show residual forgetting, the net benefit in end-to-end accuracy underscores our method's superior generalization and task adaptability.
>
> Theoretically, we posit that mitigating catastrophic forgetting and achieving strong cross-task generalization often involve inherent trade-offs, particularly in parameter-efficient continual learning setups. For instance, methods like C3 can fully avoid forgetting by dedicating resources to isolated task representations, but this comes at the expense of high training costs, scalability issues, and limited flexibility for heterogeneous schemas—making them less practical for diverse, sequential tasks like those in our streams. In contrast, K-DeCore strikes a deliberate balance: by maintaining a frozen backbone and using lightweight, trainable LoRA modules with enriched replay, we achieve robust performance across tasks while keeping computational overhead low. Our negative BWT values, though present, are markedly better than naive fine-tuning and comparable to or better than EMAR in many streams, reflecting this optimized equilibrium.
>
> To further substantiate this, our ablation studies (Table 2) show that removing key components like the dual memories leads to steeper BWT degradation, confirming their role in forgetfulness mitigation.
>
>
> ### **Weakness 2:**
> For the structure-guided query synthesis, the paper mentions that novel structures are synthesized "guided by carefully curated demonstrations" . While Appendix B is referenced for examples, more explicit details on demonstration would enhance reproducibility and understanding of this crucial step.
>
>
> ### **Response for Weakness 2:**
> Our goal is to equip the model to generalize across the combinatorial space of unseen queries for a given schema. The two principles are:
>
> 1. The demonstration set must be comprehensive by including at least one instance of every atomic structural component within the new schema. For a compositional semantic parsing schema like MTOP, this means including an example for each unique INTENT and SLOT type. For a Text-to-SQL/SPARQL schema, this includes each unique table/column and structural clause (e.g., ORDER BY, LIMIT, GROUP BY). This ensures the model has seen every basic building block it might need to use.
>
> 2. The demonstration set must be minimal yet powerful by illustrating how atomic components can be combined. Instead of providing examples for every possible combination, we provide examples of simple structures and slightly more complex ones, teaching the model the rules of composition. The model is expected to learn to generate novel, more complex structures by recombining these observed primitives.
>
>
> Two synthetic examples of MTOP and CombWebQ are shown below.
>
> MTOP:
> ```text
> Structure 1:
> [INTENT_1 [SLOT_1 VALUE_1 ] [SLOT_2 VALUE_2 ]]
>
> Structure 2:
> [INTENT_1 [SLOT_1 VALUE_1 ]]
>
> Synthetic Structure:
> [INTENT_1 [SLOT_1 VALUE_1 ] [SLOT_2 VALUE_2 ] [SLOT_3 VALUE_3 ]]
> ```
>
> MTOP:
> ```text
> Structure 1:
> SELECT DISTINCT ?x
> WHERE {
> ?x [COL_1] [ENT_1] .
> ?x [COL_2] ?y .
> ?y [COL_3] ?z .
> }
>
>
> Structure 2:
> SELECT DISTINCT ?x
> WHERE {
> ?x [COL_1] [ENT_1] .
> ?x [COL_2] ?y .
> }
> ORDER BY ?y
> LIMIT 1
>
> Synthetic Structure:
> SELECT DISTINCT ?x
> WHERE {
> ?x [COL_1] [ENT_1] .
> ?x [COL_2] ?y .
> ?y [COL_3] ?z .
> }
> ORDER BY ?z
> LIMIT 1
> ```

---

> > ### Comment · Reviewer_fB8B · 2025-08-05
> >
> > Thanks for your detailed reply. Most of my concerns have been addressed and I would like to keep my score.

---

### Official Review · Reviewer_BVyM · 2025-07-02

**Clarity:** 4
**Significance:** 3
**Originality:** 3
**Rating:** 4
**Confidence:** 3

**Summary:**

This paper proposes K-DECORE, a continual learning framework for Structured Knowledge Reasoning (CSKR) tasks, which involve translating natural language questions into structured queries over heterogeneous structured knowledge sources. The central idea is a knowledge decoupling mechanism splitting the reasoning process into schema filtering and query construction. K-DECORE maintains a fixed-size set of PEFT modules, utilizing dual-perspective memory and a strategy for synthesizing pseudo-queries to improve generalization and reduce forgetting. Experimental evaluation on four benchmarks spanning diverse SKR tasks and three LLM backbones shows that K-DECORE achieves consistent gains over existing continual learning baselines, as validated through multiple metrics and ablation studies.

**Questions:**

1. Could the authors provide a more detailed distribution of different problem types within the data stream? In scenarios where there is a significant difference between the training and test distributions, or where there is a large imbalance among different SKR categories, can the method still maintain robust performance?

2. The performance drop after ablating certain components, such as Random Memory, appears to be minimal. Could the authors provide additional experiments or explanations to clarify this observation?

3. As the proposed approach introduces several modules, it may significantly increase training cost. Although Section 4.6 compares training efficiency, the paper lacks sufficient implementation details. Could the authors provide additional experiments or elaborate further on the implementation aspects related to this comparison?

**Ethical Concerns:**

["NO or VERY MINOR ethics concerns only"]

**Final Justification:**

I appreciate the authors' detailed rebuttal, which has resolved my concerns regarding the stream distribution and the effectiveness of the submodules. However, I believe the underlying significance of this work is somewhat limited, and this issue cannot be addressed through the rebuttal. Therefore, I have decided to maintain my score of 4.

**Limitations:**

See Weaknesses and Questions

**Quality:**

3

**Strengths And Weaknesses:**

**Strengths:**
1. The experimental evaluation is thorough: methods are compared across three diverse task streams using three different LLM backbones, and baselines include a strong set of rehearsal-based and PEFT-based continual learning algorithms. Ablation studies dissect methodological contributions.
2. The decoupled architecture, i.e., splitting schema filtering and query construction, offers a principled way to enable transfer and mitigate forgetting between heterogeneous tasks. The method shows a commendable level of innovation and delivers effective results.
3. The paper is well-organized and clearly written, with illustrative figures and tables that effectively support the presentation.

**Weaknesses:**
1. It seems that the main experiments do not include a breakdown of the distribution of various SKR problem types across each stream, which could be important for interpreting the results more precisely.
2. The necessity of the designed submodules in this work is not sufficiently demonstrated. In Figure 3, the performance differences between the curves are minimal. Similarly, in Table 2, K-DeCore achieves consistently the best performance only on the AA metric.

---

> ### Author Rebuttal · Authors · 2025-07-30
>
> We are very grateful to you for providing us with valuable feedback and suggestions for our paper. We will provide explanations and clarifications for each weakness and question.
>
> ### **Weakness 1 & Question 1:**
> It seems that the main experiments do not include a breakdown of the distribution of various SKR problem types across each stream, which could be important for interpreting the results more precisely.
>
> Could the authors provide a more detailed distribution of different problem types within the data stream? In scenarios where there is a significant difference between the training and test distributions, or where there is a large imbalance among different SKR categories, can the method still maintain robust performance?
>
> ### **Response to Weakness 1 & Question 1:**
>
> It is important to emphasize that the stream we constructed consists of four heterogeneous benchmarks (GrailQA, MTOP, Spider, CWQ), each of which presents a unique task order, thereby introducing different degrees of distribution shift and data imbalance:
>
> - SKR Stream 1: GrailQA → MTOP → Spider → CWQ
> - SKR Stream 2: CWQ → Spider → MTOP → GrailQA
> - SKR Stream 3: MTOP → GrailQA → CWQ → Spider
>
> Crucially, to assess true generalization in each SKR task, we ensure that the training and test sets within each task are non-IID. The degree of this challenge is quantified by the **unseen schema rate**—the proportion of schemas in the test set unseen during training. The statistics are detailed below:
>
> | task name | # of training schema | # of test schema | unseen schema rate |
> |-----------|----------------------|------------------|--------------------|
> | GrailQA   | 934                  | 278              | 74.1%              |
> | MTOP      | 63                   | 35               | 8.6%               |
> | Spider    | 92                   | 70               | 87.1%              |
> | CompWebQ  | 310                  | 256              | 49.2%              |
>
> This demanding setup, particularly with the high zero-shot rates in GrailQA and Spider, allows for a nuanced analysis of how task heterogeneity impacts catastrophic forgetting and knowledge transfer. Across all three challenging streams, our method, K-DECORE, demonstrates remarkable robustness. Its decoupling mechanism consistently yields significant performance gains, highlighting its effectiveness in complex, non-IID continual learning environments.
>
> **To further assess our model’s robustness in more realistic scenarios, we conducted an additional experiment simulating an imbalanced data stream**, a common challenge where tasks have a non-uniform number of training samples. Specifically, we configured the training set with the following sample distribution: **Spider (500), GrailQA (800), CombWebQ (300), and MTOP (600)**. The results of this experiment are presented in the table below. Here we use Qwen2.5-7b-Instruct as the backbone model.
>
> | Method           | AA on stream1 | BWT on stream1 | FWT on stream1 | AA on stream2 | BWT on stream2 | FWT on stream2 | AA on stream3 | BWT on stream3 | FWT on stream3 |
> |------------------|---------------|----------------|----------------|---------------|----------------|----------------|---------------|----------------|----------------|
> | Fine-Tuning      | 18.5          | -15.2          | 4.1            | 22.2          | -11.8          | 4.8            | 26.3          | -5.1           | 3.4            |
> | EMAR             | 28.2          | -3.1           | 5.4            | 30.7          | -1.1           | 5.1            | 28.3          | -1.6           | 3.2            |
> | Ke-DeCore (Ours) | 31.8          | -10.9          | 8.5            | 33.5          | -8.5           | 7.3            | 30.1          | -6.5           | 6.2            |
>
> As shown in the table, our method, Ke-DeCore, demonstrates clear superiority in the imbalanced data stream experiment. It achieves the highest AA on all the streams, significantly outperforming the EMAR baseline.
>
>
> We will incorporate a full analysis of these results into the camera-ready version upon acceptance.
>
>
>
>
> ### **Weakness 2 & Question 2:**
> The necessity of the designed submodules in this work is not sufficiently demonstrated. In Figure 3, the performance differences between the curves are minimal. Similarly, in Table 2, K-DeCore achieves consistently the best performance only on the AA metric.
>
> The performance drop after ablating certain components, such as Random Memory, appears to be minimal. Could the authors provide additional experiments or explanations to clarify this observation?
>
> ### **Response to Weakness 2 & Question 2:**
>
> While the performance gap between our K-DeCore and the Random Memory baseline may appear modest in the averaged results of Figure 3 and Table 2, we wish to highlight the consistency and statistical significance of this improvement.
>
> To provide a more detailed view, the table below presents the performance delta (K-DeCore AA - Random Memory AA) across three distinct task streams from Table 2.
>
> | Backbone Model                | $\Delta$ AA(%) on SKR stream1 | $\Delta$ AA(%) on SKR stream2 | $\Delta$ AA(%) on SKR stream3 | avg $\Delta$ | std $\Delta$ |
> |-------------------------------|-------------------------------|-------------------------------|-------------------------------|------|------|
> | Llama3-8B-Instruct            | 0.6                           | 3.4                           | 0.7                           | 1.6  | 1.6  |
> | Qwen2.5-7B-Instruct | 0.4                           | 0.6                           | 0.9                           | 0.6  | 0.3  |
>
> The results show that our full K-DeCore consistently outperforms the baseline in every single run. Crucially, the low standard deviation (std) demonstrates our method’s robustness against the randomness of task sequences, whereas methods like Random Memory can be more sensitive to it. This consistent, stable gain, validated across multiple permutations, underscores the reliability of our method.
>
>
> ### **Question 3:**
> As the proposed approach introduces several modules, it may significantly increase training cost. Although Section 4.6 compares training efficiency, the paper lacks sufficient implementation details. Could the authors provide additional experiments or elaborate further on the implementation aspects related to this comparison?
>
>
> ### **Response to Question 3:**
>
> To ensure a rigorous and fair comparison, all experiments were benchmarked under a consistent environment: a single NVIDIA RTX 4090 GPU, with T5 as the shared backbone model. We calculated the average training time per task by measuring the total wall-clock time from the beginning to the end of the training process for each method and then dividing by the total number of tasks. For all compared methods, we adopted the official hyperparameters specified in their respective papers.
>
> Correspondingly, during testing, we counted the average inference time of each method on each task, from the start of inference of the first sample to the end of inference of the last sample. All methods adopted the VLLM framework for acceleration.
>
> Figure 5 in the manuscript demonstrates that our method does not add significant time cost during training or inference.
>
>  We will incorporate this detailed breakdown into the appendix of our subsequent versions.

---

> > ### Comment · Reviewer_BVyM · 2025-08-05
> > **Official Comment by Reviewer BVyM**
> >
> > Thank you for the detailed responses, which have addressed my concerns well. I would like to keep my score.

---

### Official Review · Reviewer_Zwgd · 2025-07-03

**Clarity:** 2
**Significance:** 2
**Originality:** 2
**Rating:** 4
**Confidence:** 4

**Summary:**

This paper introduces K-DeCore, a new continual learning framework for Structured Knowledge Reasoning across heterogeneous tasks. By decoupling reasoning into schema filtering and query building stages and leveraging a unified schema representation, the approach aims to facilitate knowledge transfer without increasing parameter count. A dual-perspective memory mechanism—covering schema and query structure—and a structure-guided pseudo-data synthesis strategy are incorporated to boost generalization and retention. Experiments on four SKR benchmarks with multiple LLM backbones show gains across accuracy, forgetting, and forward transfer metrics compared to baselines.

**Questions:**

Please refer to Strengths And Weaknesses

**Ethical Concerns:**

["NO or VERY MINOR ethics concerns only"]

**Final Justification:**

Here is a brief summary of my reasoning:

* On Experimental Scale: My concern about the limited sample sizes was alleviated. The authors clarified that this setup aligns with existing benchmarks and, more importantly, provided data on the "unseen schema rate".

* On Schema Mapping: The rebuttal provided a explanation of how the framework's mapping strategy handles complex, non-relational structures like graphs and nested schemas.

* On Empirical Baselines: I commend the significant effort to address the initial weakness in comparative evaluation. By implementing and evaluating additional baselines (SFNet and Apex) for decoder-only models during the rebuttal period, the authors have demonstrated their method's better performance.

Overall, the paper is better now. I have updated my final score to Borderline Accept.

**Limitations:**

Yes

**Quality:**

3

**Strengths And Weaknesses:**

Strengths

1. This paper clearly motivates that schema filtering is a common subtask across different SKR forms and proposes a systematic decoupling between schema filtering and query building.

2. Unified schema representation: By mapping various structured knowledge forms (SQL, KG, dialogue) to a DB-style schema, the model reduces task heterogeneity in schema filtering.

3. The core methodology, especially the knowledge decoupling and memory construction strategies, is clearly explained.


Weaknesses
1. All experiments constrain task train/test samples to fixed small sizes (1,000/300), which, while making experiments tractable, limits generality to much larger-scale or more realistic continual environments where distribution drift is more gradual or data is less segmented.

2. The described DB-like mapping unifies most SKR schemas, but some structured knowledge forms may not map easily (e.g., hierarchical, nested, or graph-centric schemas with types or attributes that do not naturally correspond to columns/tables).

3. The work highlights that most baselines cannot be directly ported to decoder-only LLMs, but the consequence is that comparative results with these (Llama3, Qwen2.5) have fewer baselines and thus weaker empirical competition, risking over-claiming.

---

> ### Author Rebuttal · Authors · 2025-07-30
>
> We are very grateful to you for providing us with valuable feedback and suggestions for our paper. We will provide explanations and clarifications for each weakness and question.
>
> ### **Weakness 1:**
> All experiments constrain task train/test samples to fixed small sizes (1,000/300), which, while making experiments tractable, limits generality to much larger-scale or more realistic continual environments where distribution drift is more gradual or data is less segmented.
>
>
> ### **Response to Weakness 1:**
>
> Our experimental setup aligns with established practices in existing continual learning studies, such as SAPT [1], C3 [2], and APEX [3]. For instance, SAPT employs a training set of 1,000 samples and a test set of 500. These works intentionally simulate real-world scenarios where labeled data is scarce, thereby underscoring the model’s robustness under data insufficiency—a common practical barrier in deploying continual learning systems.
>
> To further validate the generalization capability of our method, we ensured that the training and test sets for each task on the stream are not identically distributed. For example, in tasks like Spider and GrailQA, we adhered to their original splits, which inherently introduce distribution shifts (e.g., varying query schema diversities between train and test) in continual environments. The detailed information of  is shown in the following table, where the unseen schema rate indicates how many schemas in the test set do not appear in the training set.
>
> | task name | # of training schema | # of test schema | unseen schema rate |
> |-----------|----------------------|------------------|--------------------|
> | GrailQA   | 934                  | 278              | 74.1%              |
> | MTOP      | 63                   | 35               | 8.6%               |
> | Spider    | 92                   | 70               | 87.1%              |
> | CompWebQ  | 310                  | 256              | 49.2%              |
>
>
> Our results demonstrate strong performance despite these constraints, suggesting broader applicability in real continual learning scenarios.
>
>
> ### **Weakness 2:**
> The described DB-like mapping unifies most SKR schemas, but some structured knowledge forms may not map easily (e.g., hierarchical, nested, or graph-centric schemas with types or attributes that do not naturally correspond to columns/tables).
>
> ### **Response to Weakness 2:**
> **In fact, our framework is designed with the flexibility to handle such complex structures.**
>
> **1. On Graph-Centric Schemas (e.g., Knowledge Graphs):**
> Our mapping strategy has been empirically validated on complex, graph-centric schemas. We represent entities as distinct **tables** and their relations/attributes as **columns**. This approach effectively linearizes graph structures while preserving their core semantics. Our strong performance on KG-intensive benchmarks like **GrailQA** and **CompWebQ** directly demonstrates that this mapping retains the necessary information for complex, multi-hop reasoning.
>
> To ensure no loss of critical graph properties, our mapping explicitly encodes:
> *   **Directionality:** We preserve the `subject-predicate-object` structure inherent in triples. The subject (e.g., an entity) is mapped to the primary key of its table, while its outgoing relations (predicates) and their corresponding objects are mapped to distinct columns. This preserves the directed nature of the edges, preventing ambiguity during query generation.
> *   **Edge Types/Attributes:** Type information is embedded directly into the schema at the column level. For instance, a relation is prefixed with its type (e.g., `location:born_in`), creating a dedicated, typed column. This schema-level encoding allows the model to leverage type information without any modification to the underlying architecture.
>
> **2. On Hierarchical or Nested Schemas:**
> For hierarchical and nested structures, our framework is designed to accommodate established **schema flattening** techniques [1, 2]. This is a standard preprocessing step that linearizes nested data by recursively expanding multi-level fields into single-level columns (e.g., `user.address.city` becomes a column `user_address_city`). Since this transformation occurs before the data is presented to our model, it does not alter our core continual learning mechanism.
>
> We will add this part to the appendix in the subsequent version.
>
>
> ### **Weakness 3:**
> The work highlights that most baselines cannot be directly ported to decoder-only LLMs, but the consequence is that comparative results with these (Llama3, Qwen2.5) have fewer baselines and thus weaker empirical competition, risking over-claiming.
>
> ### **Response to Weakness 3:**
> Our initial baseline selection was guided by a significant technical consideration: many prior state-of-the-art methods [1,3] are intrinsically designed for the T5 encoder-decoder architecture. Their core mechanisms, such as specialized schema encoders or constrained decoding logic, are non-trivial to adapt to modern decoder-only LLMs without substantial re-engineering, which could compromise the integrity of the original methods. Therefore, we prioritized baselines that are either natively compatible with or have established adaptations for decoder-only models to ensure a fair and architecturally consistent evaluation.
>
> However, to directly address the reviewer’s concern and further strengthen our empirical claims, we have dedicated significant effort during the rebuttal period to implement and evaluate additional strong baselines on our target LLMs. Concretely, we implement SFNet[4] (2023) and Apex[3] (2025), which are relatively new and have good performance, with LLM backbones. The experimental results are shown in the following table.
>
>
> | Method | AA on stream1 | BWT on stream1 | FWT on stream1 | AA on stream2 | BWT on stream2 | FWT on stream2 | AA on stream3 | BWT on stream3 | FWT on stream3 |
> | :--- | :--- | :--- | :--- | :--- | :--- | :--- | :--- | :--- | :--- |
> | SFNet + Llama3-8B-Instruct | 36.1 | -14.5 | 3.7 | 35.8 | -17.1 | 4.2 | 35.4 | -12.3 | 4.2 |
> | Apex + Llama3-8B-Instruct | 39.2 | -11.2 | 2.1 | 40.5 | -12.6 | 3.2 | 33.4 | -12.7 | 3.6 |
> | Ke-DeCore + Llama3-8B-Instruct | 40.5 | -16.7 | 5.9 | 41.1 | -17.1 | 6.1 | 37.0 | -19.3 | 4.2 |
> | SFNet + Qwen2.5-7B-Instruct | 40.3 | -10.5 | 4.5 | 38.5 | -13.2 | 4.8 | 34.1 | -15.0 | 5.1 |
> | Apex + Qwen2.5-7B-Instruct | 41.8 | -7.5 | 3.9 | 39.6 | -8.9 | 4.1 | 35.5 | -14.2 | 4.8 |
> | Ke-DeCore + Qwen2.5-7B-Instruct | 43.2 | -8.2 | 6.9 | 40.1 | -9.8 | 5.5 | 36.8 | -16.7 | 6.9 |
>
>
> As these new results demonstrate, our method maintains a significant performance advantage over a now-broader set of competitive baselines with LLM backbones.  We will incorporate these new results and discussion in the final version of the paper.
>
>
> - [1] Zhao, Weixiang, et al. "SAPT: A Shared Attention Framework for Parameter-Efficient Continual Learning of Large Language Models." Proceedings of the 62nd Annual Meeting of the Association for Computational Linguistics (Volume 1: Long Papers). 2024.
> - [2]. Chen, Yongrui, et al. "Parameterizing context: Unleashing the power of parameter-efficient fine-tuning and in-context tuning for continual table semantic parsing." Advances in neural information processing systems 36 (2023): 17795-17810.
> - [3] Liu, Ruiheng, et al. "Filling memory gaps: Enhancing continual semantic parsing via sql syntax variance-guided llms without real data replay." Proceedings of the AAAI Conference on Artificial Intelligence. Vol. 39. No. 23. 2025.
> - [4] Chen, Yongrui, et al. "Learn from yesterday: A semi-supervised continual learning method for supervision-limited text-to-sql task streams." Proceedings of the AAAI Conference on Artificial Intelligence. Vol. 37. No. 11. 2023.

---

> > ### Comment · Reviewer_Zwgd · 2025-08-04
> >
> > Thank you for the detailed response. Several of my concerns have been addressed, and I will raise my score accordingly.

---

### Note · Authors · 2025-08-13

We sincerely thank the Area Chair and all reviewers for their insightful feedback and constructive engagement during the discussion period.

We are greatly encouraged that the reviewers unanimously recognized the core contributions of our work. They highlighted K-DECORE as a pioneering effort in Continual Structured Knowledge Reasoning (CSKR) across heterogeneous tasks, and praised our central idea of decoupling reasoning into schema filtering and query building as innovative, principled, and effective. The reviewers also valued our fixed-parameter approach for its scalability and the thoroughness of our experimental validation across multiple backbones and benchmarks.

The discussion period was invaluable for clarifying key aspects of our methodology and results. We appreciate the reviewers' positive reception of our rebuttals, with all four confirming that their primary concerns were addressed. The feedback has helped us sharpen our paper's message, particularly regarding the addition of LLM comparison methods and the clarification of the description of the data set distribution are discussed, the scope of our schema unification, and the nuances of our pseudo-data synthesis.

We believe the paper, strengthened by this dialogue, makes a solid and timely contribution to the community. Thank you once again for your time and effort in improving our work. We hope K-DECORE meets the standards for acceptance.

---

### Decision · Program_Chairs · 2025-09-17

**Decision:**

Accept (poster)

**Comment:**

K-DeCore introduces a novel continual learning framework for structured knowledge reasoning tasks that decouples reasoning into task-specific and task-agnostic stages to facilitate knowledge transfer.  By mapping various structured knowledge forms (SQL, KG, dialogue) to a DB-style schema, the method reduces task heterogeneity in the task-specific schema filtering stage.  The framework incorporates a dual-perspective memory consolidation mechanism and a structure-guided pseudo-data synthesis strategy to enhance model generalization.  Experiments on four benchmark datasets and  large language models demonstrate K-DeCore's superior performance over existing continual learning methods across multiple metrics.

In the discussion period, the authors addressed the various concerns raised by the reivewers, including the small experimental size and the applicability to complex, non-relational structures, while the reviewers mostly feel possitve about the discussion.